# Guillain–Barré Syndrome in Older People—A Case Report and Literature Review

**DOI:** 10.3390/diseases13090306

**Published:** 2025-09-18

**Authors:** Xiaomei Chen, Win Ko, Fiza Waseem, Lidia Cilcic, Nahian Kazi, Ahmed Abdelhafiz

**Affiliations:** Department of Geriatric Medicine, Rotherham General Hospital, Moorgate Road, Rotherham S60 2UD, UK; xiaomei.chen@nhs.net (X.C.); winthiri.ko@nhs.net (W.K.); fiza.waseem@nhs.net (F.W.); l.cilcic@nhs.net (L.C.); kazi.nahian@nhs.net (N.K.)

**Keywords:** Guillain–Barré syndrome, older people, clinical presentation, flaccid paralysis, autonomic dysfunction, diagnosis, management, prognosis, outcome

## Abstract

Guillain–Barré syndrome (GBS) is the most common acute inflammatory motor polyneuropathy. It affects all age groups, but the incidence increases with increasing age. Before manifesting with neurological symptoms, it is usually preceded by a prodromal phase of infection, most commonly respiratory or gastrointestinal. Pathologically, it is a post-infection immune disorder. The immune response is due to mimicry between the infecting agent and axolemmal surface molecules, which triggers an acute immune injury that leads to blockade of nerve conduction. Age-related decline in immune function plays a role in the increased prevalence and severity of GBS in older people. Typical neurological manifestations are ascending paralysis, areflexia and cranial nerve involvement, but sensory loss is uncommon. In up to 25% of cases, autonomic dysfunction occurs, which includes cardiovascular, sudomotor, gastrointestinal or genitourinary symptoms. The development of autonomic dysfunction in GBS is associated with poor prognosis. We report a case of a 78-year-old man who presented with a predominant autonomic dysfunction that led to a delay in the diagnosis. Because of the multiple morbidities associated with old age, the diagnosis of GBS presentation with autonomic dysfunction, without the typical neurological clinical pattern, may be attributed to the existing comorbidities. Therefore, clinical suspicion and close monitoring with respect to the development of autonomic dysfunction, from the first day of hospital admission, are important. The main diagnostic tests are cerebrospinal fluid analysis looking for protein–cell dissociation and nerve conduction studies to confirm the neuropathy. Treatment involves general supportive care, specific immunological intervention by intravenous immunoglobulins or plasma exchange courses and neurorehabilitation. Severe cases may require intensive care admission and mechanical ventilation. More than 80% of cases will fully recover, but 10% may have residual disability, with mortality estimated at 3–7%. Older age, multiple morbidities, severe weakness, autonomic dysfunction and the need for mechanical ventilation are poor prognostic factors.

## 1. Introduction

Guillain–Barré syndrome (GBS) is a common acute inflammatory polyneuropathy, which leads to motor weakness and flaccid paralysis [1]. GBS typically presents with ascending paralysis and sensory signs that start in the lower limbs and progress upwards. It ascends to the trunk, chest muscles and upper limbs, reaching up to the level of the cranial nerves, but clinical presentation is variable [2]. Most cases of GBS occur after a preceding infection, suggesting that it is a post-infectious, immune-mediated disorder; however, some cases occur spontaneously [3]. Globally, GBS affects up to 2.7/100,000 person-years across all age groups [1]. Although GBS appears to peak in the fourth and the sixth decades of life, the incidence generally increases with increasing age [4]. The outcome of GBS is favourable in most cases; however, the prognosis is likely to be worse in older people. It has been shown that older age is associated with prolonged length of hospital stay and hospitalisation-related complications [5]. With increasing life expectancy, the incidence of GBS in older people is set to increase. However, there is little literature on the clinical presentations and outcomes of GBS in people ≥ 75 years old, as these patients tend to present atypically with respect to infections [6,7]. Therefore, we present a case and a literature review of published cases in the English language to draw attention to the diagnostic difficulties and the outcomes of GBS in this age group.

## 2. Case Report

A 78-year-old male presented to the emergency department with a one-day history of general decline and recurrent falls. He complained of weak legs and inability to bear weight. He had a history of cough and fever during the previous 3 weeks. His past medical history included surgically removed rectal cancer with ileostomy, heart failure, stage 4 chronic kidney disease (CKD) and hypertension. Previously, he was independently mobile. Observations showed a temperature of 36.1 °C, a pulse of 70 bpm, a respiratory rate of 17/minute, blood pressure of 173/80 mmHg and oxygen saturation of 99% on room air. On examination, he had restricted bilateral shoulder and hip flexion due to pain and weakness. Muscle tone, sensation, cranial nerves, bladder and bowel control were normal. Laboratory tests revealed mild leukocytosis (white cell count at 10.40 × 10^9^/L, reference range 3.7–10.0), mild anaemia (haemoglobin at 123 g/L, reference range 132–169), neutrophilia (neutrophils at 9.5 × 10^9^/L, reference range 1.7–6.6), lymphocytopenia (0.5 × 10^9^/L, reference range 1.0–3.0), mildly raised C-reactive protein (CRP) (38 mg/L, reference range < 5.0) and stable stage 4 CKD (creatinine at 185 μmol/L, reference range 53–97). Electrocardiogram, chest X-ray (CXR) and CT brain were unremarkable. There was no specific clinical diagnosis made at this stage and he was admitted to hospital for observation. On the same day of admission, he developed multiple episodes of large amounts of coffee ground vomiting without abdominal pain. The CT abdomen and pelvis indicated oesophageal and gastric dilatation and suspicion of gastric outlet obstruction. (Figure 1) However, gastroscopy showed Barrett’s oesophagus and reflux oesophagitis but no bleeding or obstruction. Paralytic ileus was the clinical impression, although electrolytes were normal. He was treated conservatively with intravenous fluids and proton pump inhibitors (PPIs). On the fifth day of admission, he deteriorated significantly, with a new oxygen requirement of 15 L/min. He became dyspnoeic and developed a high systolic blood pressure (SBP) of 200 mmHg. Repeated CXR showed right-sided lung collapse with small right-sided pleural effusion and arterial blood gases (ABGs), demonstrating type 1 respiratory failure.

He was admitted to the high-dependency unit (HDU) for continuous positive airway pressure (CPAP) ventilation. Clinical impression was exacerbation of heart failure due to high SBP and lower respiratory tract infection due to a possible aspiration. He was treated with glyceryl trinitrate infusion, intravenous diuretics, antibiotics and nasogastric tube (NGT) feeding. The airway clearance was significantly compromised by his weak coughing effort and high sputum load despite continuous chest physiotherapy support. On the eighth day of admission, the cause of his persistent respiratory failure was revisited and clinical suspicion of a neuromuscular condition occurred. A repeat neurological examination revealed flaccid weakness in the upper limbs (power of 4/5, Medical Research Council (MRC) grade) and lower limbs (2/5 on hip flexion, knee flexion and ankle flexion bilaterally). Areflexia was noted in all limbs and plantar responses were downward moving on both sides. Sensation and cranial nerves were not affected. Lumbar puncture revealed raised protein (1.17 g/L, reference range 0.1–0.4) and normal white cell count (3 × 10^6^/L, reference range < 5). Autoimmune screen including anti-GD1a IgG Ab, anti-GD1a IgM Ab, anti-GD1b IgG Ab, anti-GD1b IgM Ab, anti-GQ1B IgG Ab, anti-GQ1B IgM Ab, anti-GM1 ganglioside IgG Ab, anti-GM1 ganglioside IgM Ab, acetyl choline receptor Ab and antinuclear Ab was negative. Screen for HIV was negative. Nerve conduction studies (NCSs) were unavailable, and he was diagnosed as GBS based on the clinical course and the cerebrospinal fluid (CSF) findings. Treatment with a 5-day course of intravenous immunoglobulins (IVIGs) was commenced. The patient gradually improved after the IVIG course and was weaned off the non-invasive ventilation and moved from HDU to a medical ward on the seventeenth day of admission. Nasogastric tube feeding continued along with neurorehabilitation. Over the next 2 weeks, he resumed satisfactory oral intake and was able to sit out or stand up with assistance. After a total of 4 weeks of hospitalisation, he was discharged to a rehabilitation facility.

## 3. Discussion

We present a case of a 78-year-old man with a clinical diagnosis of GBS. Globally, GBS remains the most common cause of acute neuromuscular paralysis. It is more common in older people, with a peak age between 50 and 70 years, and is slightly higher in men (male to female ratio of 1.5:1) [1]. Studies suggest that the incidence ranges between 0.8 to 1.9 cases per 100,000 people per year. This increases with advancing age, from 0.6 cases per 100,000 in children to 2.7 cases per 100,000 per year in elderly people aged ≥80 years [8]. In addition to old age, multiple morbidities may increase the risk of GBS. It has been shown that morbidities such as leukaemia, lymphoma, diabetes, liver disease, myocardial infarction, congestive heart failure and cerebrovascular disease are associated with a 1.6- to 4.6-fold increased risk of GBS. Furthermore, newly diagnosed morbidities and recent surgery are other factors increasing the risk {odds ratio (OR) of 4.1, 95% confidence interval (CI), 3.0 to 5.6 and 3.7, 2.6 to 5.2}, respectively [9]. GBS commonly follows a preceding infection or immune stimulation that leads to an aberrant autoimmune response, which targets the peripheral nerves and their spinal roots [10]. As a result, GBS may have seasonal fluctuations or geographical variations affecting its incidence, presumably related to variations in the incidences of the preceding infections [11,12]. Most of the preceding prodromal illnesses are gastrointestinal or respiratory bacterial infections. However, other pathogens, such as Epstein–Barr virus (EBV), cytomegalovirus (CMV), hepatitis E virus (HEV), mycoplasma pneumoniae, haemophilus influenzae, influenza A virus and Zika virus, are also involved [13]. Borrelia burgdorferi, the cause of Lyme disease, has been reported as a predisposing infection to GBS, although it is not a common cause [14].

We have summarised case reports of GBS in older people ≥ 75 years of age published in the English language over the last 10 years [15,16,17,18,19,20,21,22,23,24,25,26,27,28,29,30,31,32,33,34,35,36,37,38,39] (Table 1). Vaccine-induced GBS cases were not included due to the large volume of literature. A total of 26 cases are included, with the average age being 80 years (range 75–90); 16 (58%) cases are men, with a male-to-female ratio of 1.4:1, consistent with what is reported in the literature. Most cases (15, or 58%) had a preceding infection. Respiratory tract infection was reported in eight (31%) cases, gastrointestinal tract infections in six (23%) cases and urinary tract infection in one (4%) case. The remaining 11 cases (42%) did not recall prior infections. This is consistent with the literature, as about two-thirds of GBS cases have a history of predisposing infection [40]. The prodromal phase from infection to presentation with neurological symptoms ranged from a few days to 2 weeks [15,16,17,18,19,20,21,22,23,24,25,26,27,28,29,30,31,32,33,34,35,36,37,38,39]. In our case, the prior chest infection occurred three weeks before presentation. Typically, infections preceding GBS occur within four weeks of the onset of neurological symptoms [40]. Three cases reported having surgery prior to their symptoms. The risk of developing GBS has been reported to be higher during the first 6 weeks after surgery [41].

The underlying pathogenesis of GBS is likely related to immune injury. The main phenotypes are acute inflammatory demyelinating polyneuropathy (AIDP) and acute motor axonal neuropathy (AMAN). In the first phenotype, the immune injury occurs at the myelin sheath and Schwann cell components, while in the second phenotype, it occurs at the membranes on the nerve axon (axolemma) [42]. Although T-cell-mediated injury occurs, GBS is largely a humorally-mediated disease [43]. The antibody-mediated injury is driven by molecular mimicry between microbial and axolemmal surface molecules. The molecular mimics are glycans expressed on the lipooligosaccharides of prior infectious agents, such as Campylobacter Jejuni, which is capable of inducing antibody responses to these carbohydrate antigens [44]. The injury occurs due to complement fixing, macrophage recruitment and immune complex deposition in axolemmal membrane. This leads to disruption of the anatomical and physiological integrity of the affected nerves. As a result, blockade of nerve conduction and progression to widespread axonal degeneration occurs [45]. The immunological pathogenesis involved in the demyelinating polyneuropathy is less well understood compared with the axonal neuropathy.

This is likely due to the lack of identification of the causative antigen in demyelinating neuropathy because of the wider range of immune stimulants beyond infective organisms. Glycolipids expressed in the myelin sheath can be the primary antigen in AIDP; however, this needs to be explored in future studies [46]. GBS is not always dichotomous, and intermediate cases, which cannot be classified into one category or another, have been reported. In these cases, simultaneous inflammatory injury of glial and axonal membranes occurs [47]. In COVID-19-associated GBS, several heterogeneous mechanisms have been suggested, such as cytokine storm, secondary hypercoagulability and direct neurotropism of the virus. The clinical characteristics and disease evolution appeared to be similar to those observed in GBS secondary to other infections. Therefore, the main mechanism is likely to be post-infection dysregulation of the immune system generated by the infecting agent [48]. With increasing age, the changes in the immune cell function and age-related decline in reparative processes such as delayed de-differentiation of Schwann cells and decline in phagocytic ability of macrophages may play a role in the increased prevalence and severity of GBS in older people [49].

The classic clinical features of GBS include acute progressive flaccid paralysis of the limbs associated with decreased or absent deep tendon reflexes. Sensory loss is less common. There may be associated autonomic disturbance. The symptoms progress over the first and second weeks of presentation but rarely extend beyond four weeks [2]. The classic symptoms of GBS are summarised in Box 1. Our patient, although presented initially with leg weakness, developed significant gastrointestinal autonomic instability with copious vomiting and paralytic ileus, despite normal electrolytes. Shortly after, he developed cardiovascular autonomic instability with significantly high systolic blood pressure, leading to heart failure. Only with the development of respiratory failure did clinical suspicion of GBS occur, on the eighth day of admission. None of the cases reported in Table 1 presented with early autonomic disturbance. The main autonomic manifestations of GBS are summarised in Box 2. It has been reported that about 25% of cases may have signs or symptoms of autonomic dysfunction such as ileus, excessive sweating, blood pressure instability or cardiac arrhythmias, which may necessitate a pacemaker insertion [10]. It can present early in the acute phase of GBS or later on recovery. Although its mechanism is not well known, it involves both sympathetic and parasympathetic systems [50]. Autonomic dysfunction involving the cardiovascular system may cause significant morbidity and lead to mortality. Sudden death has been reported in up to 5–7% of severe cases [51]. Increased levels of catecholamines and impairment of baroreceptor response may explain the autonomic dysfunction, especially if preceded by cardiac surgery [52]. Older age {odds ratio (OR) 1.11, *p*  =  0.001}, hypertension (OR 0.13, *p*  =  0.028), alcoholism (OR 0.17, *p*  =  0.037) and the Guillain–Barré disability scale (GBDS) (OR 1.65, *p*  =  0.037) are risk factors for autonomic dysfunction [53]. A 1-year increase in age and a one-point increase in the GBDS increase the odds of autonomic dysfunction by 10% and 65%, respectively. A history of hypertension or increased alcohol intake decreased the odds by 88% and 83%, respectively [54]. Age-associated autonomic dysfunction is likely a contributing factor to GBS autonomic disturbance in heart rate, peripheral vascular resistance and cardiac output [54,55] The protective effect of pre-morbid hypertension could be related to the intake of anti-hypertensive medications, which may offset some of the effects of autonomic dysfunction, but the protective effects of alcohol remain unexplained and need further study. Other reported clinical predictors of autonomic dysfunction include quadriplegia, weakness of bulbar or neck muscles and the need for mechanical ventilation. The severity of autonomic dysfunction may be related to the severity of motor weakness [56]. Although there is no clear link between the preceding infectious agents and the development of autonomic dysfunction, Campylobacter Jejuni is the most frequently identified infectious agent that may trigger autonomic dysfunction in GBS [57].

Box 1Main symptoms of Guillain–Barré syndrome.
Generally unwellUrinary retention or constipationIncontinence of urine or stoolLeg weaknessLess control of facial musclesAutonomic neuropathyInability to climb stairsNight painBreathing difficultiesAscending paralysisRapid pulseRapid spread of sensorimotor deficitsEye movement weakness


Box 2Main autonomic dysfunction in GBS.
Cardiac
Arrhythmias such as sinus tachycardia or bradycardia
Vascular
Hypertension, hypotension, postural hypotension, facial flushing
Gastrointestinal
Ileus, constipation, diarrhoea, faecal incontinence
Genitourinary
Urinary incontinence, urine retention
SudomotorHyperhidrosis, anhidrosis, hyperthermia, hypothermia


The diagnosis of GBS should be made when a patient presents with the typical clinical features and classic pattern of symptoms. Symmetrical ascending neuropathy associated with reduced or absent deep tendon reflexes is a typical feature, supported by a history of recent infection. Sensory symptoms are minimal or absent, cranial nerves may be affected and autonomic dysfunction may occur. Albuminocytologic dissociation (high protein–normal cells) is the hallmark of a CSF finding in GBS. However, protein level can be normal initially in the first week and should not exclude the diagnosis of GBS. High cell levels should raise suspicion of other diagnoses such as infection. An NCS will show neuropathy, although it could be normal during the first few days. Mimics of GBS should be considered systematically from diseases affecting the central nervous system such as encephalitis, motor neurons such as motor neurone disease, nerve roots such as chronic inflammatory demyelinating polyneuropathy, plexuses such as diabetes-related neuropathies, peripheral nerves such as vasculitis, neuromuscular junction such as myasthenia gravis and muscles such as myopathies (Table 2). The diagnosis of these conditions will depend on their clinical picture and the relevant investigations. Most of the cases in Table 1 were diagnosed based on the CSF findings, and some have, in addition, supporting NCSs. Although a high level of clinical awareness is required, NCSs are important to confirm the diagnosis and provide evidence of polyneuropathy [58]. NCSs were not performed in our case or in five of the reported cases in Table 1, for various reasons, such as unavailability of the test or patients declining investigations. Out of the remaining 21 cases, 9 (43%) cases showed AIDP, 7 (33%) showed AMAN and 5 (24%) showed acute motor sensory axonal neuropathy (AMSAN). The findings suggest that AIDP is the most common type of GBS-associated neuropathy, consistent with the previous literature. A previous study reported incidences of 63% AIDP, 23% AMAN and 14% AMSAN, respectively [59]. An incidence of 33% of AMAN was also previously reported in older people [60]. In our patient, NCSs were not performed as the clinical and CSF findings were sufficient to make the diagnosis. This is a limitation of our case report as it may lead to a delay in the diagnosis. However, our patient was relatively old with multiple morbidities, which may have contributed to a delay in GBS diagnosis. In a retrospective study of 140 GBS patients, comorbidity was assessed using the age-adjusted Charlson comorbidity index (ACCI). In patients with late diagnosis of GBS (>14 days), age was significantly higher, with mean (SD) of 61.8 (15.0) years compared with 49.1 (18.4) years, *p*  =  0.001, in the early (≤14 days) diagnosis group. In addition, the duration from the onset of GBS to diagnosis was longer in patients with high ACCI (log-rank test, *p*  <  0.001). The diagnosis was significantly delayed in patients with malignancy and cardiovascular diseases, suggesting that early suspicion of GBS in patients with these morbidities is required [61].

The treatment of GBS includes general supportive care and specific immunological therapy.

Supportive therapy includes treatment for any infections, maintenance of fluid balance and close monitoring of vital signs, especially respiratory function by regular measurement of vital capacity [10]. Haemodynamic monitoring of autonomic function, bowel and bladder care along with prophylactic anticoagulation therapy and pain control are required. Anticipation and early diagnosis of autonomic dysfunction with close monitoring of heart rate, cardiac rhythm, fluctuations in blood pressure and fluid status are essential to avoid sudden unobserved deterioration and premature mortality. Examples of monitoring of cardiac autonomic dysfunction include monitoring heart rate variability (HRV) and cardiovascular reflex testing such as Valsalva’s manoeuvre, deep breathing tests and baroreflex sensitivity, and an active standing 30:15 ratio, which may predict arrhythmias or blood pressure instability [53]. The active standing 30:15 ratio measures the parasympathetic system response to active standing. A comparison of slowest heart rate at 30 s to fastest heart rate at 15 s gives the 30:15 ratio. A higher ratio > 1.04 indicates preserved parasympathetic function. A ratio < 1.01 is abnormal and between 1.01–1.03 is borderline, suggesting a potential autonomic dysfunction. Several studies have demonstrated the utility of these tests. A 24 h monitoring of HRV was feasible in a cohort of children with GBS and detected autonomic dysfunction early, while standard autonomic manoeuvres were impractical [62]. In another two studies, cardiovascular reflex testing detected autonomic dysfunction in a subset of adult patients, with implications for predicting adverse outcomes [63,64]. Therefore, close monitoring should start as soon as the diagnosis is made and continued till full recovery. The management of autonomic dysfunction is largely symptomatic therapy combined with close monitoring of any exaggerated responses due to the instability. For example, sympathetic overactivity can be treated with short-acting B-blockers with gradual titration to avoid bradycardia or hypotensive crisis. Bradycardia can be treated with atropine and intravenous fluids, although vasopressors or even cardiac pacing may be required. Gastrointestinal autonomic dysfunction can be treated with procedures such as bowel decompression and bowel rest [2]. Urine retention requires catheterisation. Pain-related autonomic dysfunction can be managed by liberal analgesia. In severe cases of autonomic dysfunction, admission to an intensive care setting is needed [65].

Plasma exchange (PE) and IVIG are the main immunological therapies available for GBS. PE machines directly remove the antibodies and other potentially injurious factors from the blood stream [66]. IVIG therapy reduces antibodies and inflammatory cytokines levels and down-regulates macrophage activity and cell adhesion molecules to limit the self-harm induced by the passage of autoimmune cells through the blood–nerve barrier [67]. In a recent systematic review and meta-analysis, there was no significant difference between IVIG or PE therapies in terms of curative effect, length of hospital stay, duration of mechanical ventilation, risk of GBS relapse or risk of complications related to the treatment regimens. This suggests that both IVIG and PE have similar curative effects [68].

In patients with poor mobility due to leg weakness, IVIG is effective when started early, within the first 2 weeks of onset. The usual dose is 0.4 g/kg of body weight daily for five consecutive days. The slower administration of IVIG over five days may avoid side effects such as treatment-related fluctuations (TRF) [69]. Five sessions of PE over 2 weeks are recommended in patients with poor mobility and these should be given in the first two weeks of presentation of muscle weakness [70]. Patients usually require either IVIG or PE, as a combination of both does not seem to be better than either alone [71]. In most cases, IVIG is tried as the first-line therapy before PE is considered. This is because IVIG therapy is widely available, more convenient to administer and has minimal side effects. Some patients will have a prolonged immune response, which causes sustained nerve damage leading to TRF. TRF usually occurs after the eighth week of presentation and can be treated with another five-day course of IVIG [72]. In patients who continue to deteriorate after the second course of IVIG, another diagnosis, such as acute presentation of chronic inflammatory demyelinating polyneuropathy, should be considered, especially in those presenting after the eighth week of the initial diagnosis of GBS [73]. Most of the cases in Table 1 were treated with IVIG and some had PE as well. Methylprednisolone was given in one case, [32] although there is no evidence to suggest the benefits of steroids in GBS [74]. In our case, there was a good response to a five-day course of IVIG. In addition to supportive and specific treatment, close monitoring is required for early identification of patients who may require mechanical ventilation or intensive care for severe autonomic instability. Eleven (42%) cases described in Table 1 required mechanical ventilation. It is estimated that around 20–30% of patients with GBS will require mechanical ventilation [8]. The higher percentage reported in Table 1 reflects the older age and the severity of symptoms in older people. Patient criteria for consideration of escalation of therapy to intensive care are summarised in Box 3. Predictors of prolonged ventilation and the need for tracheostomy include the inability to lift the arms from bed after one week of intubation and an axonal subtype or unexcitable nerves in the NCS [75].

Box 3Warning signs in GBS when considering escalation to intensive care admission. FVC: forced vital capacity.Prognostic tool:Erasmus GBS Respiratory Insufficiency Score (EGRIS) > 4.Clinical indicators:
Increasing respiratory distress

➢
*Breathless at rest*
➢
*Shallow and rapid breathing*
➢
*Reduced breath sounds at lung bases*
➢
*Paradoxical breathing*
➢
*Episodic use of accessory respiratory muscles*
➢
*Inability to count >10 in one breath*


Imminent respiratory failureSymptomatic arrhythmiasUnstable blood pressureImpaired cough reflexDifficulty swallowingRapid progression of motor weaknessBulbar palsyFacial weaknessStaccato speech with only a few words in one breathMental clouding or somnolence
Laboratory indicators
Low oxygen saturation < 92%, PO2 < 8 KP, CO2 > 6 KPFVC < 20 mL/kg body weightReduction in FVC by >30% from baselineFalling trend of FVCInconsistent FVC


Neurorehabilitation should start early in conjunction with medical therapy. The clinical outcome of GBS recovery can be measured by scales such as the MRC sum score or GBDS, which measures the functional motor disability using a score from 0 (normal) to 6 (death) [76].

Generally, GBS is a treatable condition, with up to 80% of cases fully recovering and being able to walk independently, and 50% returning to their baseline functional state [77]. Severe disability can occur in up to 10% of cases and mortality is estimated at 3–7% due to complications such as sepsis, acute respiratory distress syndrome, pulmonary emboli or cardiac arrest [77,78]. Severe disability as a result of GBS is more common in older people. In one study, GBDS > 3 and mortality were reported to be more common in people aged 60–80 compared to those < 60 years, with *p* < 0.0001 [4]. Comorbidity was present in 100% of older compared to 66% of younger group.

Out of the cases reported in Table 1, nine (35%) cases fully recovered, seven (27%) cases partially recovered, six (23%) cases remained dependent and four (15%) cases died. The higher mortality rate in these reported cases is likely due to the old age of this cohort. It has been shown that older patients have more severe disease at admission and encounter worse prognosis at the three-month follow-up. One study showed that, in comparison to younger patients, older patients had a longer duration of hospitalisation and poorer short-term prognosis. Older age (≥60 years) and lower nadir MRC scores were predictors of poor short-term prognosis in severe GBS patients [5]. In a retrospective study, AIDP was the most common neuropathy in people aged ≥ 60 years. Compared to younger adults, the elderly cohort had a shorter duration of symptoms, higher frequency of facial palsy, lower frequency of pain, lower mean MRC sum score and worse GBDS at study entry and discharge (*p* < 0.05). Requirement for mechanical ventilation and cardiac autonomic dysfunction was higher among the elderly (*p* = 0.02) [79]. Another study reported that age ≥ 70 years, rapid onset of weakness (less than 7 days from symptom onset to hospital admission), severe weakness on admission, the need for mechanical ventilation and axonal subtype of neuropathy were identified as poor prognostic markers with respect to the three-month outcome [80]. Very old age patients are likely to need more detailed attention than younger people, with comprehensive geriatric assessments (CGAs) conducted on admission. The oldest patient reported in Table 1 is a 90-year-old woman with underlying morbidities and frailty. The authors recommended early involvement of a geriatrician in her care to improve the outcome, as doing so may support acute therapies by managing other geriatric syndromes such as malnutrition, mobility problems, delirium and pressure ulcers [28].

## 4. Conclusions

We present a case of GBS in a 78-year-old man. The initial clinical presentation was dominated by autonomic dysfunction leading to paralytic ileus, with normal electrolytes, gastric dilatation and significant vomiting. Clinical suspicion of GBS occurred later, on the eighth day of admission. Therefore, GBS may be overlooked and diagnosis is delayed in older people, if it is not suspected and considered in the differential diagnosis early on following presentation. Autonomic dysfunction is not normally the dominant feature of GBS and occurs in only 25% of cases. Because of multimorbidities associated with old age, early occurrence of autonomic dysfunction, without the classic symptoms of GBS, may be misdiagnosed as related to the pre-existing conditions. This may result in a delay in the diagnosis and initiation of treatment. Older age and multimorbidities are also associated with poor outcomes, which underscores the importance of early diagnosis and treatment of GBS. Therefore, clinicians should be aware of, and consider, GBS in the differential diagnosis of patients presenting with non-specific symptoms, especially those related to autonomic dysfunction. Furthermore, close monitoring of cardiovascular instability and gastrointestinal and genitourinary functions, from the first day of admission, is vital for early detection and treatment of autonomic dysfunction.

## Figures and Tables

**Figure 1 diseases-13-00306-f001:**
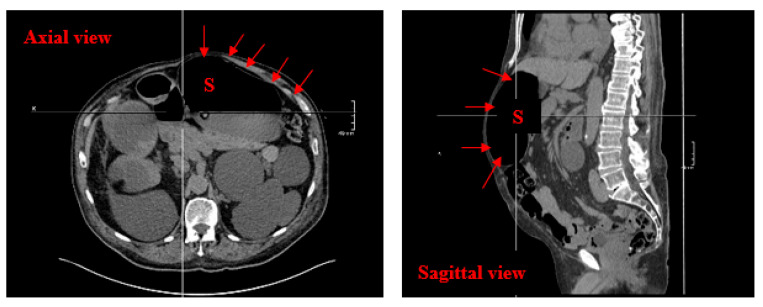
CT abdomen without contrast showing a dilated stomach without clear evidence of mechanical gastric outlet obstruction. S: stomach.

**Table 1 diseases-13-00306-t001:** Summary of GBS reported cases in older people ≥ 75 years old.

Study	Patient/Presentation	Examination	Investigations	Treatment	Outcome
Watanabe K, et al., case report, Japan, 2025 [13].	84 M, weakness in right upper limb, gastroenteritis 4 days previously.	Progression of weakness to all limbs, areflexia, ↓ touch sense.	NCS, axonal neuropathy, progressed to respiratory failure.	IVIG, mechanical ventilation, PE.	Partial recovery after rehabilitation for 187 days.
Solodovnikova Y, et al., case report, Ukraine, 2025 [14].	77 M, acute symmetric ascending flaccid tetraparesis, recent history of pneumonia.	Areflexia, glove-stocking–glove sensory loss, urine retention.	CSF consistent with meningitis, progression of neurological symptoms.	IV antibiotics, IVIG 5 days.	Partial recovery of neurological symptoms.
Gómez-Dabó L, et al., case report, Spain, 2025 [15].	75 F, progressive impairment of bilateral III, IV, VI, VII, IX, X XII cranial nerves, recent gastroenteritis.	Bilateral ophthalmoplegia, ptosis, mydriasis, facial, tongue and palate palsy, ↓ pharyngeal reflex, flaccid dysarthria, brisk tendon reflexes, all limb dysmetria.	NCS, acute inflammatory demyelinating polyneuropathy.	Mechanical ventilation, IVIG 5 days.	Partial recovery after 5 months.
Thiriveedi M, et al., case report, US, 2025 [16].	86 M, progressive generalised weakness, ↓ mobility, recent COVID-19 infection.	Weakness in all limbs, ↓ pain and vibration sense, hyporeflexia	CSF, ↑ protein 95.9 mg/L, leukocytes 1/µL.	IVIG 5 days, then a second course given, NCS, axonal demyelinating polyneuropathy.	Minimal motor recovery, discharged to care home after 24 days.
Ghishan S, et al., case report, US, 2025 [17].	83 M, fall, acute lower extremity weakness, dysphonia.	Symmetrical weakness, intact sensation, absent reflexes.	CSF, ↑ protein 179 mg/L, mild leucocytosis, progressed to respiratory failure.	Mechanical ventilation, PE.	Partial recovery.
Kota NK, et al., case report, India, 2025 [18].	81 M, acute onset bilateral lower limb weakness followed by upper limb over 1 day.	Complete areflexia with reduced power in four limbs.	NCS, motor peripheral neuropathy, CSF ↑ protein 1261.3 mg/dl, leukocytes 2/µL.	IVIG 5 days.	Full recovery in 2 weeks.
Min X, et al., case report, China, 2024 [19].	79 F, progressive limb weakness 7 days after craniotomy for cerebellar contusion.	Quadriplegia, autonomic dysfunction, dilated pupils, respiratory failure.	CSF, ↑ protein 1.3 g/L, leukocytes 3/µL.	Mechanical ventilation, IVIG 5 days.	Good recovery after 6 weeks rehabilitation.
Ito S, et al., case report, Japan, 2024 [20].	81 M, dysarthria, dysphagia, upper limb weakness.	Inability to protrude tongue beyond the dental arch, muscle weakness in distal upper limbs.	CSF, ↑ protein 48 mg/dl, leukocytes 5/mm^3^, NCS, reduced amplitudes and velocities in median and ulnar nerves.	IVIG 5 days.	Partial recovery after 31 days rehabilitation.
Chen FY, et al., Taiwan, 2024 [21].	75 F, symmetric weakness in both distal lower limbs and ataxia.	Absent deep tendon reflexes in knee and ankle, ascending weakness, constipation and urine retention.	NCS, acute inflammatory demyelinating polyradiculoneuropathy, CSF, albuminocytologic dissociation.	IVIG 5 days.	Partial recovery one month after rehabilitation.
Reddy V, et al., case report, India, 2024 [22].	79 M, cough, malaise, fever, diplopia, watery diarrhoea 5 days previously.	Left-beating nystagmus, postural tremor in both arms, intermittent leg jerking, exaggerated reflexes in biceps and patella.	CSF, ↑ protein 82 mg/dl, leukocytes 13–20/mm^3^, blood cultures campylobacter jejuni.	IV antibiotics.	Mobility improved after three months of physical therapy.
Obara K. case series, Japan, 2024 [23].	1. 81 M, diplopia, unsteady standing, URTI 3 days previously. 2. 78 M, diplopia, tingling in both arms, enterocolitis 4 days previously.	1. Dysarthria, bilateral ptosis, ataxia, weakness in upper limbs, progressed to flaccid tetraplegia, areflexia. 2. Eyes, fixed, flaccid tetraplegia.	1. NCS, consistent with axonal neuropathy with demyelinating features., CSF, ↑ protein 1216 mg/dl, leukocytes 6/mm^3^ 2. NCS, axonal neuropathy with demyelinating features, CSF, albuminocytologic dissociation.	1. Mechanical ventilated and intubation, PE and IVIG. 2. Mechanical ventilation and intubation, PE and IVIG.	Only slight recovery with residual weakness. 2. Only slight recovery with residual weakness.
Wen PY, et al., case report, China, 2023 [24].	75 F, sub-acute upper limb weakness, progressed to lower limbs, recent myocardial infarction.	Hypotonia in limbs, quadriplegia, hypoesthesia, bulbar palsy, tendon areflexia.	Declined investigations, progressed to respiratory failure.	Mechanical ventilation, IVIG 5 days.	Died after 25 days.
Lee J, et al., Korea, 2023 [25].	79 M, inability to move both legs, numbness in feet, weakness and tingling in both upper extremities after spinal fusion surgery.	Weak legs, decreased tendon reflexes in upper and lower extremities.	NCS, sensory–motor peripheral polyneuropathy in upper and lower limbs, CSF ↑ protein 381 mg/dl, leukocytes 96/mm^3^	IVIG 5 days.	Improved after 6 weeks, fully recovered after 1 year.
Sidoli C, et al., case report, Italy, 2023 [26].	90 F, fatigue, worsening gait and leg strength, dysphonia, dysarthria and dysphagia, COVID-19 positive previous 3 W.	Dysarthria, dysphonia, dysphagia, symmetric weakness in upper limbs, asymmetric weakness in lower limbs, asymmetric sensory signs in lower limbs, absent tendon reflexes in lower limbs, normal plantar response.	NCS acute motor and sensory axonal neuropathy, CSF ↑ protein 57 mg/dl and cell count 21/µL.	IVIG 5 days.	Poor recovery, died after 2 months.
Tu WC, et al., case report, Taiwan, 2022 [27].	87 M, weakness in lower limbs, surgery for neck injury 2 weeks previously.	Limb weakness.	CSF, ↑ protein 167 mg/dl, absent leukocytes, NCS, acute axonal polyneuropathy.	Mechanical ventilation, supportive care, rehabilitation.	Remained dependent, died after 3 years.
Luvsannyam E, et al., case report, Netherlands, 2021 [28].	83 F, bilateral numbness in lower extremities, ↓ mobility, COVID-19 positive previous few weeks.	↓ Sensations, 0/5 strength, diffuse areflexia in lower limbs.	NCS acute inflammatory demyelinating polyneuropathy, CSF, no leukocytes, ↑ protein 64 mg/dl consistent with albuminocytologic dissociation.	IVIG 5 days and three courses PE.	Improved significantly after rehabilitation.
Ramakrishna KN, case report, US, 2020 [29].	76 M, unsteadiness, poor oral intake, dysarthria, dizziness.	Respiratory distress, drowsy, impaired gag reflex, limb weakness, reduced reflexes.	NCS, acute inflammatory demyelinating polyneuropathy, CSF, ↑ protein 62 mg/dl, leukocytes < 3/mm^3^.	Mechanical ventilation, IVIG 5 days.	Tracheostomy-dependent after 6 months, mobilising with walker.
Wang Y, et al., case report, China, 2018 [30].	80 M, 10-day progressive ascending limb weakness, fever 2 weeks previously.	Weakness in limbs, areflexia.	NCS, peripheral nerve demyelination, CT lung cancer.	Methylprednisolone 80 mg QD for 10 days, then 40 mg QD for 10 days, then prednisone 10 mg QD for 3 months.	Progression, bedbound after 6 months.
Doctor GT, et al., case report, UK, 2018 [31].	81 F, confusion, fever, mild headache and photophobia, diarrhoea, vomiting 2 weeks previously.	New symmetrical limb weakness, areflexia.	CSF, ↑ protein 72 mg/dl, leukocytes 14/mm^3^, NCS, axonal polyneuropathy.	IVIG three times at 4 week intervals.	Partial recovery after 3 months rehabilitation.
Helgeson SA, et al., case report, US, 2018 [32].	81 F, weakness lower limbs, paraesthesia in feet after extubation from pneumonia.	Absent tendon reflexes, progressive weakness to upper limbs.	NCS, axonal sensorimotor peripheral neuropathy.	PE, Mechanical ventilation.	Died after 11 days.
Jo YS, et al., case report, Korea, 2018 [33].	75 F, weakness in upper and lower limbs, fever, frequency, dysuria 10 days previously.	Limb weakness and areflexia.	NCS, axonal motor type polyneuropathy.	IVIG	Almost full recovery, mild residual weakness.
Miyagi T, et al., case report, Japan, 2017 [34].	77 M, progressive dysarthria, dysphagia, bilateral upper limb weakness, diarrhoea 10 days previously.	Weak facial, tongue, cervical, pharyngeal, palatal upper limb muscles, dysphonia, dysphagia.	CSF, ↑ protein 57.4 mg/dl, normal leukocytes, NCS, reduced compound muscle action potential.	IVIG 5 days, IVIG 5 days on day 33.	Good recovery 63 days after rehabilitation.
Takahashi H, et al., case report, Japan, 2017 [35].	79 F, progressive tetraparesis, cough and fever 7 days previously.	Limbs, ocular and facial muscle weakness, reduced limb sensation.	CSF, ↑ protein 65 mg/dl, leukocytes 40/mm^3^, NCS, reduced compound muscle action potential.	IVIG 5 days.	Full recovery 37 days after rehabilitation.
Ha LD, et al., case report, US, 2017 [36].	76 M, diaphoresis, slurred speech, urinary and faecal incontinence.	Lethargic, altered mental state, bradycardia, tachypnoea, areflexia in four limbs.	CSF, ↑ protein 62 mg/dL, normal leukocytes.	PE, supportive care.	Died of multi-organ failure.
Lin YK, et al., case report, Taiwan, 2016 [37].	82 F, acute ascending weakness, distal numbness in four limbs, URTI previous 1 W.	Flaccid weakness, absent tendon reflexes in four limbs, absent plantars, ↓ peripheral sensation.	CSF protein 161 mg/dl (N = 15–45 mg/dl), leukocytes 1 (N = 0–5 cell/µL), NCS ↓ motor and sensory action potentials in all limbs, positive for SS.	Ventilation, six courses PE, IVIG for 5 days, immunosuppressant hydroxychloroquine.	Full recovery, discharged after 1 month, remained well after 1 year follow-up.

M: male; URTI: upper respiratory tract infection; W: week; NCS: nerve conduction study; CSF: cerebrospinal fluid; F: female; UTI: urinary tract infection; N: normal; IV: intravenous; IG: immunoglobulins; PE: plasma exchange; SS: Sjögren syndrome; ↑: increased; ↓: reduced.

**Table 2 diseases-13-00306-t002:** Mimics of GBS.

Mimic	Diagnostic Investigations
Central nervous system Encephalitis, encephalomyelitis, brain stem stroke, brain stem compression, leptomeningeal tumours, Wernicke’s encephalopathy, transvers myelitis.	Clinical picture, CT, MRI, CSF, serum thiamine.
Motor neurons Amyotrophic lateral sclerosis, progressive spinal muscular atrophy, poliomyelitis, West Nile virus anterior myelitis.	Clinical picture, MRI, NCS, CSF, viral serology.
Nerve roots Acute onset CIDP, autoimmune nodopathy, Lyme disease, viral-related radiculitis, leptomeningeal malignancy.	Clinical picture, NCS, CSF, serology, MRI.
Nerve plexus Diabetes mellitus, amyotrophic neuralgia.	Clinical picture, NCS, blood glucose.
Peripheral nerves Acute onset CIDP, diphtheria, vasculitis, porphyria, thiamine deficiency, electrolytes disturbance, Lyme disease, toxin-induced neuropathy, critical illness polyneuropathy.	Clinical picture, autoimmune and vasculitic screen, serology, serum electrolytes, NCS.
Neuromuscular junction Myasthenia gravis, Lambert–Eaton myathenic syndrome, botulism, organophosphorus poisoning.	Clinical picture, NCS, serum antibodies.
Muscles Hypokalaemic periodic paralysis, acute myositis, acute colchicine myopathy, critical illness myopathy.	Clinical picture, serum K, serum CK, NCS.

CSF: cerebrospinal fluid; NCS: nerve conduction studies; CIDP: chronic inflammatory demyelinating polyneuropathy; C: creatine kinase.

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
