# Peer review of "Guillain–Barré Syndrome in Older People—A Case Report and Literature Review"

_diseases, 2025, doi:10.3390/diseases13090306_

Round 1
Reviewer 1 Report
Comments and Suggestions for Authors
- Formatting of References
In-text citations do not comply with the journal’s guidelines. All references should be numbered in order of appearance and presented in square brackets (e.g., [1], [2,3], [4–6]) instead of the current format. Reference list formatting should be revised accordingly. - Clarity of Expression (Lines 39–40)
The sentence “With increasing age of the population and increased life expectancy, the incidence of GBS in this age group is likely to increase” should be reformulated for clarity and conciseness. - Aim of the Study (Lines 43–45)
The stated aim of the study could be presented more clearly. The phrasing is somewhat repetitive and should be streamlined to highlight the novelty and importance of analyzing GBS in patients ≥75 years. - Pathogen Spectrum (Lines 117–118)
While several pathogens are listed as triggers of GBS, the manuscript omits discussion of Borrelia burgdorferi (Lyme disease). Although less frequent, Lyme disease has been reported as a possible antecedent infection for GBS and should be at least acknowledged. - Tables and Boxes
Tables and text boxes should be reformatted according to journal standards. At present, they are not optimally presented and could be made clearer and more reader-friendly. - Autonomic Nervous System (ANS) Dysfunction (Lines 166–167)
The manuscript briefly mentions autonomic disturbance, but this part is left rather general. It would be useful to provide a fuller explanation, especially about the mechanisms involved. Autonomic dysfunction in GBS can present with important problems such as sudden changes in blood pressure, irregular heart rhythm, or loss of normal heart rate control, so this topic deserves more attention.
It would also add value if the authors linked these disturbances with different infections that may precede GBS, since some pathogens are thought to trigger immune reactions that affect autonomic as well as peripheral nerves.
Because autonomic dysfunction is already highlighted as a key point in the conclusion, expanding this discussion in a clearer and more detailed way would make the manuscript stronger both scientifically and clinically.
- Risk Factors and Diagnostic Modalities (Lines 182–185)
The section on risk factors for autonomic dysfunction (age, hypertension, alcoholism, disability scale) is informative, but it remains incomplete. To provide a fuller picture, the authors should expand on diagnostic modalities that have been used to assess autonomic nervous system involvement in GBS, such as:
- Heart rate variability (HRV) analysis,
- Cardiovascular reflex testing (deep breathing, Valsalva maneuver, baroreflex sensitivity),
- Continuous blood pressure and heart rate monitoring.
Importantly, the discussion would be strengthened by citing and summarizing findings from previous studies that applied these tools in GBS populations. For example, reporting how often autonomic abnormalities were detected by HRV or reflex testing, and whether these measures had prognostic or management implications, would add both scientific depth and clinical value.
General Closing Comment
Overall, this is a reasonably written manuscript that provides useful insight into GBS in patients aged ≥75 years. The inclusion of a case report is relevant, particularly given the patient’s significant comorbidities (heart failure, stage 4 chronic kidney disease). The case illustrates that GBS may easily be overlooked in patients with multiple comorbidities, underlining the clinical relevance.
The work highlights an underexplored subgroup and contributes to a better understanding of clinical presentation and outcomes in this vulnerable population. Nevertheless, the manuscript would benefit from revisions in clarity of expression, improved formatting, and a more comprehensive discussion of autonomic dysfunction and its diagnostic assessment. Addressing these issues would strengthen both the scientific and clinical impact of the paper.
Author Response
Many thanks for your comments and suggestions to improve this manuscript.
1. Formatting of References
In-text citations do not comply with the journal’s guidelines. All references should be numbered in order of appearance and presented in square brackets (e.g., [1], [2,3], [4–6]) instead of the current format. Reference list formatting should be revised accordingly.
Done
2. Clarity of Expression (Lines 39–40)
The sentence “With increasing age of the population and increased life expectancy, the incidence of GBS in this age group is likely to increase” should be reformulated for clarity and conciseness.
Shortened
3. Aim of the Study (Lines 43–45)
The stated aim of the study could be presented more clearly. The phrasing is somewhat repetitive and should be streamlined to highlight the novelty and importance of analyzing GBS in patients ≥75 years.
We removed the repetition and focused it to the point.
4. Pathogen Spectrum (Lines 117–118)
While several pathogens are listed as triggers of GBS, the manuscript omits discussion of Borrelia burgdorferi (Lyme disease). Although less frequent, Lyme disease has been reported as a possible antecedent infection for GBS and should be at least acknowledged
Thank you, added.
5. Tables and Boxes
Tables and text boxes should be reformatted according to journal standards. At present, they are not optimally presented and could be made clearer and more reader-friendly.
OK, will liaise with the editorial board for the required changes that comply with the journal style. An image version of tables/boxes will be available if required.
6. The manuscript briefly mentions autonomic disturbance, but this part is left rather general. It would be useful to provide a fuller explanation, especially about the mechanisms involved. Autonomic dysfunction in GBS can present with important problems such as sudden changes in blood pressure, irregular heart rhythm, or loss of normal heart rate control, so this topic deserves more attention
We agree, it is an important point, thank you. We have expanded in this point as suggested.
7. Risk Factors and Diagnostic Modalities (Lines 182–185)
Expanded as requested.
General Closing Comment
Thank you
Reviewer 2 Report
Comments and Suggestions for Authors
The manuscript discusses an important and current topic, Guillain-Barré syndrome (GBS) in elderly patients. The subject is relevant to neurological and clinical practice because of the unique diagnostic challenges, prognosis considerations, and the multiple comorbidities associated with this group. The article is quite well documented and includes a wealth of data from the specialized literature. However, I believe it could be improved with clearer language, better synthesis, and more emphasis on critical analysis. Here are my suggested improvements:
- The table of cases could be supplemented with a descriptive analysis of the cases and their classification into AIDP, AMAN, AMSAN, and Miller-Fisher syndrome (percentages of AIDP versus AMAN/AMSAN).
- An analysis and critical discussion of the role of comorbidities (diabetes, hypertension, and other cardiovascular diseases) in the evolution of the disease would be helpful.
- A discussion on the challenges of differential diagnosis in the elderly (including diabetic neuropathy, stroke, acute onset CIDP, and toxic or paraneoplastic neuropathies) and the importance of nerve conduction studies in reaching a final diagnosis would be helpful. Also, mention in the case presentation that the absence of NCS and EMG is a limitation of the investigation.
- I recommend further discussion about potential differences in response to IVIg compared to plasmapheresis in the elderly population, as well as the possible adverse effects of these treatments.
- Could you clarify what the GBS disability status scale measures?
- To enhance the discussion section and offer a more comprehensive view, I suggest including the following works:
- Zhang, B., Wu, X., Shen, D., & colab. (2017). The clinical characteristics and short-term prognosis in elderly patients with Guillain-Barré syndrome. Medicine (Baltimore), 96(4), e5848 (It indicates that elderly patients tend to have a more favorable short-term outlook, but with distinct clinical particularities)
- Peric S, Berisavac I, Stojiljkovic Tamas O, et al. Guillain-Barré syndrome in the elderly. J Peripher Nerv Syst. 2016;21(2):105–110
- Sidoli, C., Bruni, A. A., Beretta, S., Bellelli, G. (2023). Guillain-Barré syndrome AMSAN variant in a 90-year-old woman after COVID-19: a case report. BMC Geriatrics, 23, 114 (Exemplifies the AMSAN variant in a 90-year-old patient after COVID 19 infection, with discussions on geriatric neurological challenges)
- Stoian A, Moțățăianu A, BărcuÈ›ean L, et al. Understanding the mechanism of action of intravenous immunoglobulins: A ten-year experience in treating Guillain-Barré Syndrome. 2020;68(3):426-435. doi:10.31925/farmacia.2020.3.7.
- Stoian A, BălaÈ™a R, Grigorescu BL, et al. Guillain-Barré syndrome associated with COVID-19: A close relationship or just a coincidence? (Review). Exp Ther Med. 2021;22(3):916. doi:10.3892/etm.2021.10348 (They provide essential data on immunological mechanisms and differential diagnosis, and explain the clinical response to these types of treatment, also illustrating the most frequently encountered adverse effects)
- França Jr, M. C., et al. (2005). Guillain-Barré syndrome in the elderly: clinical, electrophysiological, therapeutic, and outcome features. Arquivos de Neuro‑Psiquiatria (Analysis addressing the frequency of axonal form in the elderly and its prognostic implications, including treatments used)
- Nagappa, M. (2017). Clinical commentary: Guillain-Barré syndrome in the elderly. (Comment article) (Brief review of clinical and evolutionary characteristics of GBS in patients over 60 years old)
- Hagen, K. M., et al. (2021). The Neuroimmunology of Guillain-Barré Syndrome and Aging. Frontiers in Aging Neuroscience (Explores age-related changes in the immune system and their relevance to GBS severity)
- Stoian A, Bajko Z, Maier S, et al. High-dose intravenous immunoglobulins as a therapeutic option in critical illness polyneuropathy accompanying SARS-CoV-2 infection: A case-based review of the literature. Exp Ther Med. 2021;22:1182. doi:10.3892/etm.2021.10616 (to be completed in Table 2 under differential diagnosis and include critical illness polyneuropathy, since without nerve conduction studies, a differential diagnosis is difficult)
Otherwise, the article is valuable but needs revision for better English language quality, increased critical analysis, and clearer organization. Rewrite Box 1 and Box 2 using more appropriate medical terminology (for example, replace "Less control of facial muscle" with "cranial nerve dysfunction" and "Incapable climbing stairs" with "motor deficit"). Explain terms like "Night pain" more clearly, and reorganize the tables for improved clarity.
Comments on the Quality of English LanguageIt can be enhanced.
Author Response
Many thanks for your comments and suggestions to improve the manuscript.
- The table of cases could be supplemented with a descriptive analysis of the cases and their classification into AIDP, AMAN, AMSAN, and Miller-Fisher syndrome (percentages of AIDP versus AMAN/AMSAN).
We have added this in text with percentages compared to literature.
- An analysis and critical discussion of the role of comorbidities (diabetes, hypertension, and other cardiovascular diseases) in the evolution of the disease would be helpful.
Added in start of discussion (as risk factors for GBS) and later as a cause of delayed diagnosis and poor outcome.
- A discussion on the challenges of differential diagnosis in the elderly (including diabetic neuropathy, stroke, acute onset CIDP, and toxic or paraneoplastic neuropathies) and the importance of nerve conduction studies in reaching a final diagnosis would be helpful. Also, mention in the case presentation that the absence of NCS and EMG is a limitation of the investigation.
Limitation added. Differential diagnosis is in Table 2 for mimics of GBS and relevant investigations.
- I recommend further discussion about potential differences in response to IVIg compared to plasmapheresis in the elderly population, as well as the possible adverse effects of these treatments.
Added.
- Could you clarify what the GBS disability status scale measures?
Added.
- To enhance the discussion section and offer a more comprehensive view, I suggest including the following works
Thank you, we have added all suggested references.
Otherwise, the article is valuable but needs revision for better English language quality, increased critical analysis, and clearer organization. Rewrite Box 1 and Box 2 using more appropriate medical terminology (for example, replace "Less control of facial muscle" with "cranial nerve dysfunction" and "Incapable climbing stairs" with "motor deficit"). Explain terms like "Night pain" more clearly, and reorganize the tables for improved clarity.
Thank you, we have used the Guillain-Barré as an acronym, that is why we followed the letters in that order. We think it will be an easy to remember take home message. However, this can be changed if still required.
English can be enhanced.
Have been reviewed throughout the manuscript.
Reviewer 3 Report
Comments and Suggestions for Authors
The abstract is not informative and not meaningful: it is mandatory to rewrite it.
It is necessary to revise the keywords, which are not significative; and to add a list of abbreviations.
The paper is full of repetitions (increasing ...preceding...due...).
Line 55: "for miles".....?
Line 70: "on the next day"...?
The discussion does not address the novelty of the case, and there is no comparison with the literature reported.
The box 1 and the table 2 are simply useless with the respect to the focus of your case.
The conclusions are so weak and inconsistent.
Comments on the Quality of English LanguageThe English must be improved: i.e. it is mandatory to cancel all the repetitions.
Author Response
Many thanks for your comments and suggestions to improve the manuscript.
The abstract is not informative and not meaningful: it is mandatory to rewrite it.
Re-written as suggested.
It is necessary to revise the keywords, which are not significative; and to add a list of abbreviations.
Reviewed. All relevant key words are included but we are open to further advice. We mentioned the abbreviations in the text but can provide a list if this is consistent with the journal style.
The paper is full of repetitions (increasing ...preceding...due...).
Line 55: "for miles".....?
Line 70: "on the next day"...?
Thanks, we have removed repetitions and focused the text across the manuscript.
The discussion does not address the novelty of the case, and there is no comparison with the literature reported.
Thanks, discussion has now been expanded and new references added.
The box 1 and the table 2 are simply useless with the respect to the focus of your case.
We have used Guillain-Barré as an acronym in Box 1 for an easy take home message and Box 2 is relevant as a differential diagnosis list for GBS. We think they are useful and complimentary as their contents are not fully mentioned in text.
The conclusions are so weak and inconsistent.
Shortened and focused.
The English must be improved: i.e. it is mandatory to cancel all the repetitions.
Thanks, we have reviewed the English and repetitions across the manuscript.
Round 2
Reviewer 1 Report
Comments and Suggestions for Authors
We sincerely thank the authors for their detailed responses to the previous set of comments. The revisions have improved the overall clarity and structure of the manuscript. However, a few additional points still require attention:
- Please ensure that a dot is placed at the end of the sentence on line 216, and provide the appropriate reference.
- The font appears inconsistent in lines 265–272. Kindly unify the formatting.
- The same issue is noted in lines 293–297; please adjust the font to match the rest of the manuscript.
- There is no need to explain the active standing test (30:15); it would be sufficient to simply mention it (lines 289–292).
- The references should be arranged properly, with numbers replacing letters in accordance with the journal’s style.
Author Response
Many thanks for your further comments:
1. Please ensure that a dot is placed at the end of the sentence on line 216, and provide the appropriate reference.
Done
2. The font appears inconsistent in lines 265–272. Kindly unify the formatting.
Corrected
3. The same issue is noted in lines 293–297; please adjust the font to match the rest of the manuscript.
Corrected
4. There is no need to explain the active standing test (30:15); it would be sufficient to simply mention it (lines 289–292)
Deleted, thanks.
5. The references should be arranged properly, with numbers replacing letters in accordance with the journal’s style
Done.
Reviewer 2 Report
Comments and Suggestions for Authors
After revising the article, I consider its structure to be markedly improved, and the manuscript ready for publication
Author Response
Thank you.
Reviewer 3 Report
Comments and Suggestions for Authors
I repeat that the keywords are not relevant: they must be revised.
I repeat that the box 1 and the table 2 are useless.
Comments on the Quality of English LanguageThe English must be improved.
Author Response
Many thanks for your further comments.
1. I repeat that the keywords are not relevant: they must be revised.
Thank you, we have added some more key words. We are open for further suggestions of any particular keywords you may think is still needed.
2. I repeat that the box 1 and the table 2 are useless
Thank you, we can always delete these, but will leave this as an editorial decision.
3. The English must be improved.
Manuscript has been reviewed.